# Antibody Properties Associate with Clinical Phenotype in LGI1 Encephalitis

**DOI:** 10.3390/cells12020282

**Published:** 2023-01-11

**Authors:** Susann Ludewig, Leonie Salzburger, Alexander Goihl, Jana Rohne, Frank Leypoldt, Daniel Bittner, Emrah Düzel, Burkhart Schraven, Dirk Reinhold, Martin Korte, Péter Körtvélyessy

**Affiliations:** 1Department of Cellular Neurobiology, Zoological Institute, 38106 Braunschweig, Germany; 2Neuroinflammation and Neurodegeneration Group, Helmholtz Centre for Infection Research, 38124 Braunschweig, Germany; 3Institute of Molecular and Clinical Immunology, Otto-von-Guericke-University Magdeburg, 39120 Magdeburg, Germany; 4Department of Neurology, Christian-Albrechts-University/University Hospital Schleswig-Holstein, 24105 Kiel, Germany; 5Neuroimmunology Unit, Institute of Clinical Chemistry, University Hospital Schleswig-Holstein Kiel/Lübeck, 24105 Kiel, Germany; 6Department of Neurology, Otto-von-Guericke-University Magdeburg, 39120 Magdeburg, Germany; 7German Center for Neurodegenerative Diseases (DZNE), 39120 Magdeburg, Germany; 8Institute for Cognitive Neurology and Dementia Research, 39120 Magdeburg, Germany; 9Health Campus Immunology, Infection and Inflammation (GC-I3), 39120 Magdeburg, Germany; 10Department of Neurology, Campus Benjamin Franklin, Charité-Universitätsmedizin Berlin, Corporate Member of Freie Universität Berlin and Humboldt-Universität zu Berlin, 12200 Berlin, Germany

**Keywords:** LGI1 encephalitis, autoimmune encephalitis, pathophysiology encephalitis, synaptic plasticity, epitope of anti-LGI1 antibodies

## Abstract

Autoimmune encephalitis (AE) associated with autoantibodies against leucine-rich glioma-inactivated protein-1 (LGI1) can present with faciobrachial dystonic seizures (FBDS) and/or limbic encephalitis (LE). The reasons for this heterogeneity in phenotypes are unclear. We performed autoantibody (abs) characterization per patient, two patients suffering from LE and two from FBDS, using isolated antibodies specified with single amino acid epitope mapping. Electrophysiological slice recordings were conducted alongside spine density measurements, postsynaptic Alpha-amino-3-hydoxy-5-methyl-4-isoaxole-proprionate-receptors (AMPA-R) and N-methyl-D-aspartate-receptors receptor (NMDA-R) cluster counting. These results were correlated with the symptoms of each patient. While LGI1 abs from LE patients mainly interacted with the Leucine-rich repeat section of LGI1, abs from both FBDS patients also recognized the Epitempin section as well. Six-hour incubation of mouse hippocampal slices with LE patients autoantibodies but not from the FBDS patients resulted in a significant decline in long-term potentiation (*p* = 0.0015) or short-term plasticity at CA3-CA1 neurons and in decreased hippocampal synaptic density. Cluster differentiation showed no decrease in postsynaptic AMPA-R and NMDA-R. LGI1 autoantibodies selected by phenotype show an almost distinct epitope pattern, elicit disparate functional effects on hippocampal neurons, and cause divergent effects on spine density. This data illuminates potential pathomechanisms for disease heterogeneity in LGI1 AE.

## 1. Introduction

Autoimmune-mediated encephalitis (AE) associated with autoantibodies (abs) against leucine-rich glioma-inactivated protein 1 (LGI1) is the second most frequent form of AE [1,2]. Patients often initially experience frequent episodes of mostly unilateral myoclonic jerks of the face and arm referred to as faciobrachial dystonic seizures (FBDS). If left untreated, patients can develop limbic encephalitis (LE) with cognitive dysfunction, focal and generalized epileptic seizures, and behavioral changes. Initiating immunosuppressive therapy during the FBDS stage of anti-LGI1-encephalitis mostly results in a good outcome without residual cognitive dysfunction [3]. Delayed treatment in patients with LE is associated with a less favorable outcome with a relevant fraction of patients suffering from lasting cognitive dysfunction [1,3].

LGI1 is a secreted protein mainly expressed in the Cornu Ammonis (CA) 1 and 3 section of the hippocampus and the Dentate Gyrus [4,5]. They are composed of two main sections: a leucine-rich repeat section (LRR) being responsible for LGI1/LGI1 dimerization and an epitempin domain (EPTP) mediating the LGI1/ADAM22/23 interaction, thus forming a protein complex [6,7]. This complex is crucial for synaptic transmission by orchestrating presynaptic K_v_1.1 channels and postsynaptic α-amino-3-hydroxy-5-methyl-4-isoxazole-propionate receptor (AMPA-R) [8,9,10].

Serum from LGI1 AE patients interrupts the hippocampal LGI1-ADAM22/23 interaction and leads to a decrease in AMPA-R density, thus modifying synaptic transmission [10,11]. This observation was confirmed in a murine animal model [12] in which chronic infusion of LGI1 ab-enriched CSF resulted in a decrease of AMPA-R and K_v_1.1 channels. Differentially detected epitopes against the LRR or EPTP domains—interfering with LGI1-LGI1 dimerization, LGI1-ADAM22/23 interaction or both—were suspected as a possible explanation for the different clinical symptoms [13], but experiments with a simple epitope screening using monoclonal abs directed against each epitope did not confirm that hypothesis [14]. They also proposed that LGI1 abs are only pathogenic when preventing the LGI1 protein from binding to the ADAM22/23.

We hypothesize that, maybe, different antibody properties and LGI1 antibody epitopes are responsible for the phenotypic variation mirrored by different electrophysiological results. We applied a translational approach with in-depth characterization of individual anti-LGI1 abs in a pilot trial with four patients suffering from LGI1-associated AE.

## 2. Materials and Methods

Every patient gave written and informed consent. This study is approved by the ethics committee of the medical faculty of the University Magdeburg in 2016 (100/16). Anonymized data will be shared by request from any qualified investigator.

### 2.1. Autoantibody Screening and Purification

Indirect immunofluorescence tests were performed for each patient on commercially available antigen-specific transfected HEK293 cells from serum and CSF (EUROIMMUN, Lübeck, Germany). No other AE-associated ab including Contactin-associated protein-like 2 (Caspr2) abs, N-methyl-D-aspartate receptor (NMDA-R) abs, AMPA-R, and Gamma-Aminobutyrate receptor nor onconeural abs (EUROIMMUN, Lübeck, Germany) were found in the sera or CSF of these patients.

### 2.2. Epitope Determination

The LGI1 sequence (#NP_005088.1) without the N-terminal signal peptide was elongated by neutral GSGSGSG linkers at the C- and N-terminus to avoid truncated peptides (PEPperPRINT GmbH, Heidelberg, Germany). The elongated antigen sequence was translated into 7, 10 and 13 amino acid (aa) peptides with peptide-–peptide overlaps of 6, 9 and 12 aa. Peptides were cyclized via thioester linkage to achieve conformational epitopes resulting in 1584 different peptides, which were dotted on a microarray and framed with HA control peptides (YPYDVPDYAG). Fluorescence intensity of LGI1 ab binding from Sera was detected as a 16-bit grayscale picture with an LI-COR Odyssey Imaging System (Bad Homburg, Germany) for each patient. Results were analyzed with PepSlide^®^ Analyzer software (Version 2.0 SICASYS, Heidelberg, Germany).

### 2.3. Determination of IgG Subclasses of Anti-LGI1 ab Fractions and Purification

HEK293 cells were transfected with the LGI1 gene containing pCMV3-SP-N-His (Biozol, Eching, Germany). LGI1 was purified from harvested supernatants of LGI1-producing clones by Ni-chelate chromatography and a subsequent anion exchange chromatography using an ÄKTA-FPLC (GE Healthcare, Freiburg, Germany). Purity was confirmed by SDS-PAGE with Coomassie staining. LGI1 was bound to a 1 mL HiTrap^®^ NHS-activated HP column (GE Healthcare) according to the manufacturer’s guide. For ab isolation, the sera were loaded overnight onto the column. Then, the column was washed with 15 mL 20 mM Tris (pH 7.5) and 15 mL 0.5 M NaCl in 20 mM Tris (pH 7.5). Anti-LGI1 ab were eluted with 100 mM glycine (pH 2.2) with ensuing neutralization using 1 M Tris (pH 8.8). In total, 2 µg of each patient-derived ab were dotted on a nitrocellulose membrane (GE Healthcare). Membranes were probed with mouse anti-human IgG1-POD, anti-human IgG2-POD, anti-human IgG3-POD, and IgG4-POD ab (ThermoFisher, Dreieich, Germany), and sheep-anti-human IgG (Seramun, Heidesee, Germany).

### 2.4. HLA-Mapping

HLA-class II-typing was performed using a commercially available sequence-specific oligonucleotide-PCR-technique at the Institute of Transfusion Medicine, University Hospital Schleswig-Holstein Kiel/Lübeck, Germany (Table 1).

### 2.5. Isolation of IgG Fractions

In total, 750 µL serum or plasma samples from patients or healthy controls were mixed with 450 µL protein A/G agarose (SantaCruz, Heidelberg, Germany). IgG fractions were isolated according to the manufacturer’s instructions. Purity was checked by Coomassie staining. Protein concentration was determined using Pierce^®^ BCA Protein Assay Kit (ThermoFisher).

### 2.6. Electrophysiology

In vitro extracellular recordings were performed on acute hippocampal slices of 4 to 9 month old C57BL/J6 or thy-1 mEGFP (on a C57BL/6J-SV129 background) mice. Slice preparation was performed as described in [15]. Slices were continuously incubated with either IgG of the four different patients or from healthy controls in 10 mL gently carbogenated artificial cerebrospinal fluid (ACSF) containing 4.5 µg/mL IgG preparations including 1.5 h recovery after preparation in a custom made incubation chamber at room temperature. We used a submerged recording setup where 30 mL of carbogenated ACSF with the respective ab circulated in a closed loop system with a flow rate of 1.2–1.5 mL/min during the entire experiment at 32 ± 0.2 °C. Field excitatory postsynaptic potentials (fEPSPs) were recorded at the Schaffer collateral to CA1 pathway as described before [15,16]. After a baseline recording of 10 min, abs were applied for another 20 min basal recording. Afterwards, long-term potentiation (LTP) was induced by application of Theta-Burst stimulation (TBS: 10 trains of 4 pulses at 100 Hz in a 200 ms interval, repeated 3 times). The Input–Output measurements were performed with defined current values (25–250 µA). Presynaptic function was assessed with the paired-pulse facilitation (PPF) paradigm by applying a pair of two closely spaced stimuli in inter-stimulus intervals (ISI) ranging from 10 ms to 160 ms before and after the LTP measurements.

### 2.7. Slice Preparation

Next, 400 µm thick transversal hippocampal slices of thy-1 mEGFP mice of electrophysiological experiments were fixed with 4% PFA overnight, dehydrated in 30% sucrose in PBS, cut into 30 µm thick slices with a freezing microtome (Frigomobil, Science Services, Munich, Germany) and mounted with Fluoro Gel (EMS, Hatfield, PA, USA). Immunohistochemistry was performed on free floating slices (20–30 µm).

### 2.8. Imaging and Analysis of Spine Density in Hippocampal Slices

Imaging of second- or third-order apical dendritic branches of hippocampal pyramidal neurons of area CA1, CA3 and DG was performed with an Axioplan 2 imaging microscope (Zeiss, Oberkochen, Germany) using a 63× oil objective under fluorescence light of 475 nm. The number of spines were determined per micrometer of dendritic length (in total 100 μm) using ImageJ (1.48v, National Institutes of Health, Bethesda, MD, USA). A minimum of three animals and five neurons per animal were analyzed, blinded to treatment.

### 2.9. Immunohistochemistry/Cluster Differentiation

Determination of postsynaptic AMPA-R levels was performed on 20 µm thick brain slices. Sections were permeabilized with 0.1% TritonX-100 in PBS (3× 5 min), saturated in PBS with 1% TritonX-100 containing 5% donkey serum and 5% BSA for 60 min and incubated with polyclonal guinea pig anti-AMPA-R1 (#AGP009, 1:200, Alomone, Jerusalem, Israel) and chicken anti-Synapsin (#106006, 1:1000, SySy, Göttingen, Germany) in 1% BSA /1% donkey serum in 0.3% TritonX-100 in PBS overnight at 4 °C. The secondary antibodies included donkey anti-guinea pig-Cy3 and donkey anti-chicken-Cy5 (#706-166-148 and #703-175-155, 1:500, 2 h, Dianova, Hamburg, Germany). For NMDA-R immunofluorescence, an antigen retrieval with pepsin (R2283, 20 min 37 °C, Sigma-Aldrich, Taufkirchen, Germany) was performed before permeabilizing the sections with 0.1% TritonX-100/PBS (3 × 5 min). Following saturation with 0.1% TritionX-100 in PBS containing 5% donkey and 5% goat serum for 60 min, primary antibodies of mouse anti-NR2B glutamate receptor (clone N59/20, #75-097, 1:1000, NeuroMab, Davis, CA, USA) and chicken anti-Synapsin 1–2 in 1% donkey and 1% goat serum in PBS with 0.1% TritonX-100 were applied for 48 h at 4 °C. In case of Caspase 3 staining, no antigen retrieval was performed, only permeabilization with 0.2% TritonX-100/PBS (3 × 5 min) and blocking using 5% rabbit/5% donkey serum in 1× PBS0.2%TritonX-100 following incubation with Anti-Caspase 3 (active form, #AB3623, 0.5 mg/mL, Millipore, Darmstadt, Germany) overnight at 4 °C. Secondary antibodies included goat anti-mouse IgG2a, donkey anti-chicken-Cy5 (1:500, 2 h; Dianova, #115-165-206 or Dianova, #703-175-155) and donkey anti-rabbit-Cy5 (1:500, 2 h; Dianova, #711-175-152). The AMPA-Rs and NMDA-Rs that were collocated with the presynaptic marker synapsin were defined as synaptic. Total and synaptic clusters of indicated protein were visualized and quantified by confocal laser scanning imaging (40× oil objective (NA 1.3), BX61WI FluoView 1000, Olympus, Hamburg, Germany) using Imaris suite 9.3.0 (Bitplane, Zürich, Switzerland).

### 2.10. Data Analysis and Statistics

Data of electrophysiological recordings were analyzed with LABVIEW software (Version 4.7, National Instruments, Austin, TX, USA). The entire statistical analysis was performed using GraphPad Prism (Version 7.05 GraphPad, San Diego, CA, USA). Data were compared using an unpaired two-tailed Student’s *t*-test. Values of **p* < 0.05 were considered significant. All data are indicated as mean ± SEM. 

## 3. Results

### 3.1. Patients Description

Patient FBDS#1 had pilo-arrective seizures and FBDS, and patient FBDS#2 had only FBDS. Both FBDS patients did not show signs of cognitive impairment and had a normal performance in a recurrent CERAD plus test. Patient LE#1 was admitted with LE after a series of four grand mal seizures on one day. MRI showed a T2/FLAIR lesion in the right hippocampus. Investigation on the previous symptoms revealed no signs of FBDS and no complex partial seizure before admission. Patient LE#2 had a series of complex partial seizures and a history of cognitive decline and was classified as LGI1-LE. Immunosuppressive therapy via methylprednisolone (MP), plasma exchange and rituximab were initiated in patient LE#1 and patient FBDS#1. Patient FBDS#2 had a complete recovery after recurrent MP. Patient LE#2 had MP and rituximab and did not recover completely, with residual subjective amnestic deficits, as did patient LE#1.

Serial MRI showed a subsequent hippocampal sclerosis (Schelten’s scale 3) in both LE-patients. Patients FBDS#1 and #2 had no hippocampal sclerosis upon follow-up MRIs.

### 3.2. IgG Subclass Analysis and Epitope Mapping Reveal Differences between Patients with LE and FBDS

All patients’ LGI1-specific autoantibodies were predominantly IgG4; however, only the LE patients and FBDS#2 had a considerable amount of LGI1-IgG3 (Figure 1A). IgG1 and IgG2 were also detected in all samples from all patients.

LGI1 abs of patient LE#1 bound two epitopes with high affinity, one within the LRR section and the other within the dimerization motif. Epitopes of LGI1 abs from patient LE#2 were detected in the LRR-CT region and in the outer part of β-sheet#1 of the EPTP In contrast, LGI1 abs of patients FBDS#1 and #2 detected epitopes mainly in the first and fourth repeat of the EPTP section, which is important for binding to ADAM22/23. Antibodies isolated from patient FBDS#2 bound an additional epitope at the transition of both sections. (Figure 1B and Appendix A).

### 3.3. Electrophysiological Effects of LGI1 abs Differ between Patients with FBDS and the Patients with LE

We incubated acute hippocampal slices with each IgG fraction. A one hour incubation with an individual IgG fraction did not alter the activity-dependent synaptic plasticity in any patient (Figure 2A,A’,A’’). In contrast, a longer incubation time of 6 h led to reduced long-term potentiation (LTP) only in slices treated with the IgG of patient LE#1 (111.67 ± 4.07% vs. 126.29 ± 3.94%, *p* = 0.015, Figure 2C,D). Incubation of hippocampal slices over a time period of 6 h with IgG fractions of patient FBDS#1 (*p* = 0.45), #2 (*p* = 0.98) or patient LE#2 (*p* = 0.601) did not affect LTP (Figure 2A’–D’’’ and Table 1).

Application of the respective ab-containing solutions did not change the basal synaptic transmission at the CA3-CA1 neurons (Figure 3) neither after 1 h (Figure 3B,B’,B’’, nor 6 h (Figure 3C,C’,C’’,C’’’) of incubation with patient IgG, indicating normal postsynaptic function. Although the IgGs of patient LE#1 altered long-term potentiation after 6 h, the IgGs of patient LE#2 impaired short-term plasticity after 6 h (Figure 3G’’’). The abs of patients suffering of FBDS did not alter presynaptic function after 1 h or 6 h of incubation (Figure 3D’–G’,D’’–G’’). 

### 3.4. Hippocampal Spine Density Is Reduced by IgG from the Patients with LE Compared to IgG from Patients with FBDS

We evaluated spine density as a correlate of excitatory synapses. Incubation with individual patient IgG fractions over a time period of 1 h did not influence the spine density in all hippocampal regions analyzed. Incubation with patient LE#1 and #2 IgG fractions for 6 h caused a significant reduction in spine density in *Dentate Gyrus* granule cells (Figure 4C, *p* = 0.014; Figure 4C’’’, *p* < 0.0001). Spine density at CA3 neurons, only analyzed for patient LE#1 and #2, were only significantly reduced following 1 h IgG application of patient LE#2 at CA3 basal dendrites (Appendix A, *p* = 0.0142) and after 6 h IgG patient LE#2 treatment at both dendritic compartments (Appendix A, *p*(apical) = 0.040; *p*(basal) < 0.0001), while the CA1 neurons remained largely unaffected despite significant reduction upon 6 h of treatment with IgG fractions of patient LE#2 at the CA1 apical dendrites (Figure 4B’’’, *p* = 0.0152). The spine numbers detected at the neurons upon 1 and 6 h of treatment with IgG fractions of patient FBDS#1 or #2 were comparable to healthy controls in the DG and CA1.

### 3.5. No Significant Changes of Postsynaptic AMPA-R Density Observed in Slices after Treatment with LGI1 ab from Patient LE#1, but Increased Cell Apoptosis after LGI1 ab Treatment from Patient LE#2

First, we confirmed LGI1 ab binding in hippocampal slices after different incubation times. In the *Stratum Radiatum* of CA1 and the upper DG we recognized a weak signal after 1 h incubation with IgG fractions of patient LE#1 (Figure 5A,A’), while it increased well after 6 h of incubation time. IgG fractions of patient LE#2 show accumulation in cells (Figure 5A’,B’ indicated by arrows) that increased with incubation time. The control IgG staining was comparable to slices that remained for 6 h in ACSF (Figure 5B’,B’’). Next, we looked whether the IgG fraction accumulation correlates with apoptosis and stained for Caspase 3. Indeed, we found a strong co-localization of IgG fractions of patient LE#2 with Caspase 3 in hippocampal slices, but not upon treatment with IgGs of healthy controls or in ACSF perfused slices (Appendix A).

Then, cluster analysis was performed in the two LE patients with significant changes seen before to assess if total and synaptic levels of AMPA-R and NMDA-R as well as the number of presynapses were altered upon abs incubation (Figure 6). Neither after 1 h nor 6 h were the amount of synapsin clusters changed in second- or third-order apical dendritic branches of CA1 neurons (Figure 6D,H). Further, synaptic AMPA-R and NMDA-R were not affected (Figure 6A,C,E–H), while only total AMPA-R clusters were slightly enhanced after incubation over a time period of 6 h with IgG fractions of patient LE#1 (Figure 6A,B).

## 4. Discussion

Our data can offer a more precise look at the pathomechanisms of LGI1 AE. Probably, the localization of the epitope does explain the diverging symptoms in LGI1 AE. The more precise mapping shows that the access from outside may also play a role and not only whether the EPTP or LRR is somewhere affected. 

In LGI1 AE, the clinical stage has a major impact on the outcome and, therefore, should have different pathomechanisms underneath. The good clinical outcome at the stage of FBDS is well in-line with our not significant changes seen in the electrophysiological recordings and the absence of morphological alterations. The poor outcome of patients having LE correlates well with the impaired electrophysiological properties as well as with the decline in spine density. 

In former experiments, using serum from LGI1 AE patients without correction for protein levels and not purified for IgG, Okhawa and coworker observed a decreasing effect on AMPA-R [11]. LGI1 abs from patients induced apoptosis-promoting proteins in hippocampal neurons in cell cultures [17]. In a recent study by Petit-Pedrol and coworker [13], who used IgG fractions only isolated by protein A/G beads but for a longer exposure over days, a reduction in presynaptic K_v_1.1 and postsynaptic AMPA-R was seen. We used a more time constrained protocol within six hours, and observed impaired long-term synaptic plasticity mediated by the NMDA-R at the Schaffer collateral-CA1 pathway [18] for the LE-patient LE#1, and impaired short-term plasticity for patient LE#2. In contrast to the authors mentioned above, we found no alterations in basal synaptic transmission mediated by AMPA-R in our “acute” approach. Maybe the changes seen in the work by Petit-Pedrol are due to a longer exposure to abs than 6 h, as occurred in our setup. Analysis of the AMPA- and NMDA-R clusters indicated that neither the amount of presynapses nor that of the receptors inserted into the postsynaptic membrane were changed following the LTP recording after 1 h or 6 h incubation with the LGI1 IgG’s of patient LE#1. Only a change in total AMPA-R was noticed, which did not contribute to the significant LTP impairment as the synaptic AMPA-Rs were unchanged between the different conditions. However, after detecting the pronounced decrease in spine density in the hippocampal slices we saw increased cell death following treatment with the LGI1 IgG’s of patient LE#2, which might explain the result.

The biggest difference to the other published paper is the downscaled epitope mapping per single amino acid [13]. This allows us to obtain a more precise insight into the possible mechanisms of action than before. In general, when projecting the epitopes onto the LGI1 protein, we see that patients with LE are having epitopes not only in the EPTP or at the Linker region, but especially, considering the tertiary structure, these epitopes must be easily accessible from the outside (Figure 1B). This easy access is also a possible explanation for the significant effects on the electrophysiological and synapse density measurements after six hours, as seen with our LE LGI1 abs. Ramberger et al. found antibodies directed against both epitopes in 28 out of 31 patients, and 31 out of 31 with EPTP epitopes again. The LGI1 IgG’s of patients FBDS#1 and #2 contained anti-LGI1 abs, which bind to epitopes interfering with the binding site of LGI1 to ADAM22. In contrast, the LGI1 IgG’s of patient LE#1 were lacking these specific anti-LGI1 epitopes. Ramberger et al. proposed one possible mechanism of action in LGI1 AE, which was that the LGI1 abs bind to the LGI1 protein before it binds to ADAM22/23 [14]. We do think that our data support this hypothesis, at least in three-quarters of our patients. Other authors [11,13] performed the epitope mapping with domain-constructs and found almost only anti-LGI1 abs, which bind epitopes in both major domains: EPTP and LRR. We did not confirm this finding in total with our more detailed epitope mapping methods using peptides, with one LE patient having no epitope at the EPTP domain. This raises the question whether epitope spreading may be one mechanism behind the evolution of symptoms in LGI1-associated encephalitis as seen in other autoimmune diseases [19].

## 5. Conclusions

Although our study comprised of four LGI1 patients, our results should be able to raise the hypothesis that the combination of anti-LGI1 IgG3 and epitopes aiming at localizations are easily accessible, especially at the LRR section, which resulted in the significant effects in our experiments. In further studies, a higher number of patients as well as other experimental setups are needed to evaluate the anti-LGI1 IgG subclasses contribution and epitope spreading to anti-LGI1 ab-mediated AE over the course of the disease.

## Figures and Tables

**Figure 1 cells-12-00282-f001:**
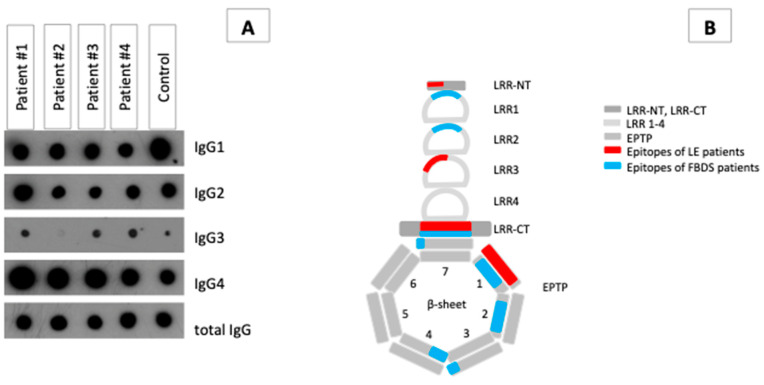
Ab properties on four anti-LGI1 ab-associated AE patients. (**A**) Dot blot analysis splitting up the LGI1-specific IgG fraction of the four patients into IgG subclasses. (**B**) Schematic presentation of the data of the epitope mapping, which show the binding of the anti-LGI1 abs of the respective patients to the LGI1 molecule according to a tertiary structure model of Yamagata and coworker [7]. Blue colors represent epitopes of FBDS patients, and red represent epitopes of LE patients.

**Figure 2 cells-12-00282-f002:**
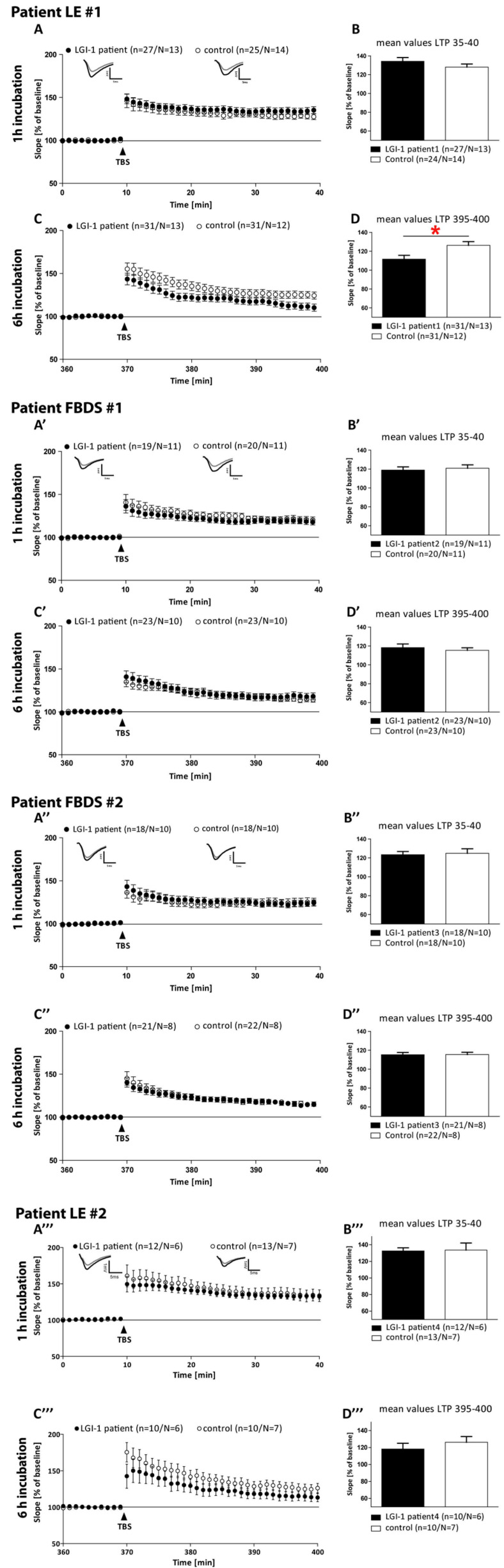
Field excitatory postsynaptic potentials (fEPSPs) of acute hippocampal slices of C57Bl/6-GFP mice treated with control serum (open circles) or with anti-LGI1- autoantibodies of four different patients (black circles) were recorded in the CA1 region by stimulating the Schaffer collateral axons of area CA3 at a frequency of 0.1 Hz. LTP was induced by the application of TBS after a 10 min baseline recording (arrowhead) in slices treated for 1 h or 6 h with control or anti-LGI1 ab of the respective patient. The LTP induction rate is shown as a percentage % of mean baseline slope. During the LTP measurement, slices were continuously perfused with either control or anti-LGI1 abs. (**A**,**A’**,**A’’**,**A’’’**) LTP recordings of slices treated with abs of individual patients for 1 h. (**B**,**B’**,**B’’**,**B’’’**) Bar graph depicting the averaged potentiation values of the last 5 min of individual LTP recordings. (**C**,**C’**,**C’’**,**C’’’**) LTP of acute slices continuously perfused for 6 h with individual patient abs. (**D**,**D’**,**D’’**,**D’’’**) Quantification of the last 5 min of the LTP recording after long-term ab application. All data are indicated as the mean ± SEM, analyzed by student’s *t*-test. * *p* < 0.05, n = number of recorded slices, N = number of animals; significant results are marked with a red asterisk.

**Figure 3 cells-12-00282-f003:**
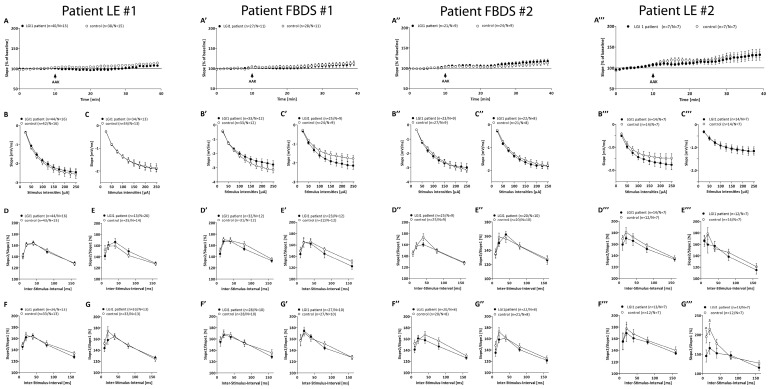
Additional electrophysiological recordings. Field excitatory postsynaptic potentials (fEPSPs) of acute hippocampal slices of C57Bl/6-GFP mice treated with control (open circles) or with LGI1 IgGs of each patient (black circles, frequency of 0.1 Hz). (**A**–**A’’’**) fEPSP recording with 40% of maximum slope for 10 min. Continuous wash-in of LGI1 IgGs did not alter the fEPSP signal. Input–Output curves revealed no differences in postsynaptic function after 1 h (**B**–**B’’’**) and additionally after 6 h (**C**–**C’’’**) treatment of acute slices with either control or LGI1 IgG. PPF paradigm before (**D**–**D’’’**) and after (**E’’’**) the first LTP recording revealed no differences in presynaptic function after 1 h treatment of acute slices with either control or LGI1 IgGs. PPF paradigm before (**F**–**F’’’**) and after (**G**–**G’’’**) the second LTP recording after 6 h of ab treatment revealed no differences, despite a significantly lower PPF curve following 6 h of LGI1 IgGs treatment of patient LE#2 after the LTP measurement ((**G’’’**), *p*(10 ms) = 0.04, *p*(20 ms) = 0.04).

**Figure 4 cells-12-00282-f004:**
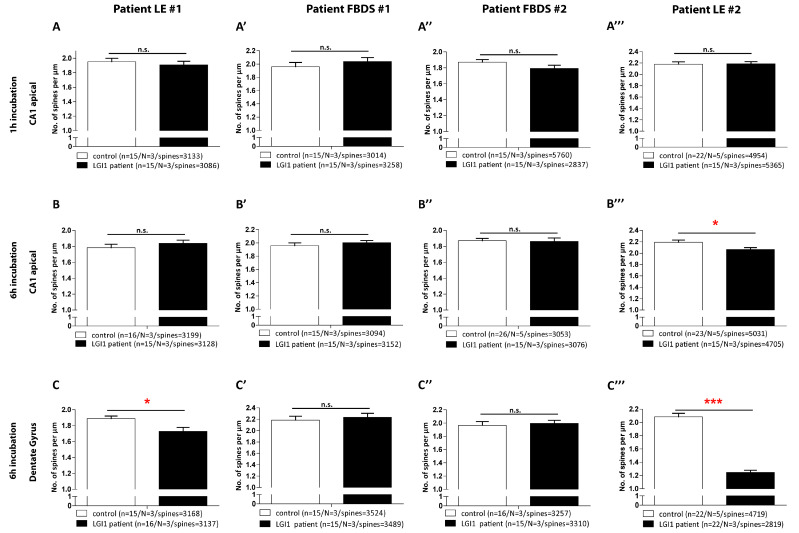
Quantified spine density after short and long ab application in CA1 and DG. Comparable spine density at apical second or third-order dendritic branches of CA1 pyramidal cells after 1 h (**A**,**A’**,**A’’**,**A’’’**) application of individual LGI1 IgG and 6 h (**B**,**B’**,**B’’**,**B’’’**). Only 6 h treatment with IgGs of patient LE#2 reduced spine density (*p* = 0.015) (**C**,**C’**,**C’’**,**C’’’**). Spine numbers at granular dendrites after 6 h incubation with individual IgG caused a reduction in spine density for patient LE#1 (*p* = 0.014 and #4 *p* < 0.001). Data presented as mean ± SEM, n = number of dendrites, N = number of animals, spines = number of total spines. Significant results are marked with a red asterisk. * = *p* < 0.05, *** = *p* < 0.01, n.s. = not significant.

**Figure 5 cells-12-00282-f005:**
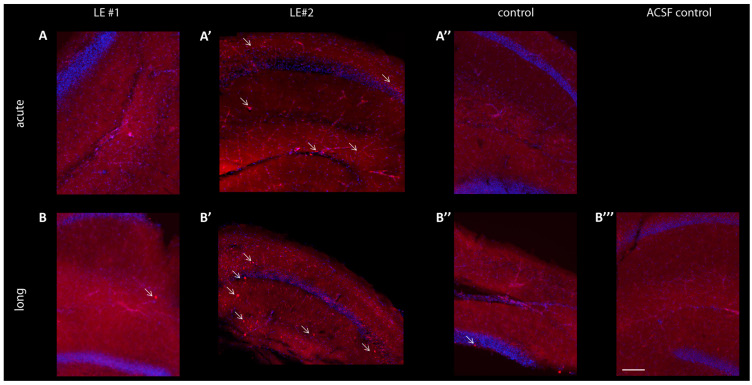
Reactivity of autoimmune antibodies (AB) on hippocampal slices. Representative images of *Cornu Ammonis* area 1 (CA1) and *Dentate Gyrus* (DG) of mouse hippocampal slices. Live hippocampal slices were incubated with individual patient or control sera (4.5 µg/mL). Hippocampal slice with anti-human IgG staining (red) after 1 h (**A**) of ab incubation and after 6 h (**B**). Staining of human IgGs following 1 h (**A’**) or 6 h (**B’**) live incubation with patient control serum (red). (**B’’**) Hippocampal slice remaining for 6 h in ACSF. Scale bar: 200 μm.

**Figure 6 cells-12-00282-f006:**
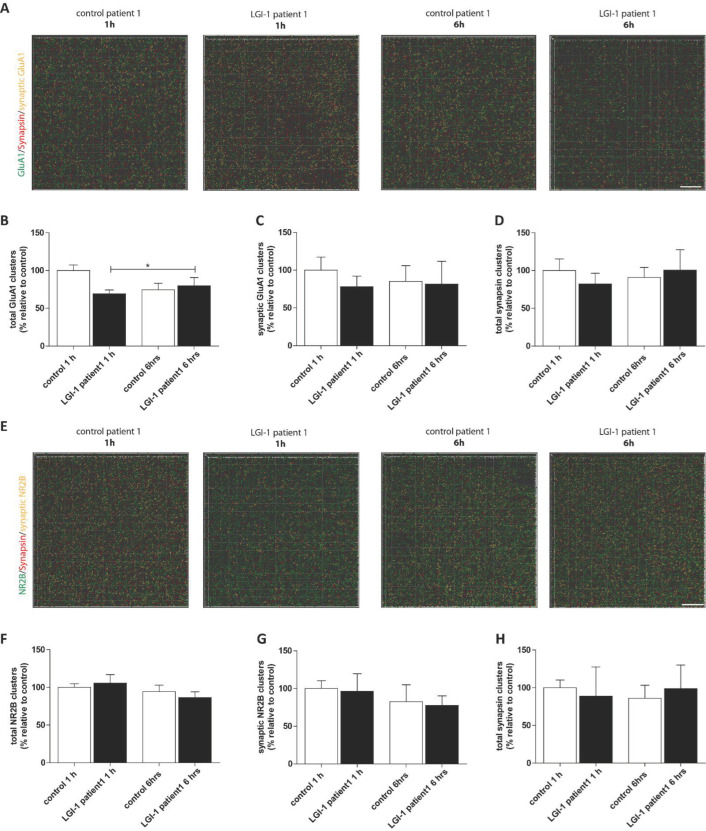
Cluster differentiation of NMDA-R and AMPA-R. Patient LE#1-derived LGI1 abs cause no alteration in synaptic AMPA-R or NMDA-R ratios. (**A**,**E**) A 3D projection of analyzed AMPA-R (upper panel) and NMDA-R (lower panel) clusters in second to third-order dendritic branches of area CA1 that were colocalized or non-colocalized with synapsin. Scale bar = 5 µm. Quantification of the total (**B**) and synaptic (**C**) AMPA-R clusters, and synapsin clusters (**D**) in hippocampal acute slices treated for 1 or 6 h with patient-derived LGI1 or control abs after the LTP recording. Quantification of the total (**F**) and synaptic NMDA-R clusters (**G**), and synapsin clusters (**H**) in hippocampal acute slices of individual mice treated for 1 h or 6 h with patient-derived LGI1 or control abs after the LTP recording. The mean density in 1 h control ab treated slices was defined as 100%. Data are represented as the mean ± SEM and analyzed by one-way ANOVA followed by Turkeys post-hoc test stating * *p* < 0.05.

**Table 1 cells-12-00282-t001:** Synopsis of patients’ characteristics and the results from the electrophysiological recordings, spine density measurements, epitope mapping. Significant results are written in red. (Data presented as mean ± SEM, *N* = number of animals, *n* = number of recorded slices or analyzed dendrites, *p*-value as a result of unpaired Students *t*-test, *p* < 0.05).

Recorded data	Variable	LE #1	LE #2	FBDS #1	FBDS #2
Symptoms		Limbic encephalitis	Limbic encephalitis	FBDS, complex partial sz.	FBDS
LGI1 titre	initial	1:100	1:20	1:1000	1:160
	after treament	1:30	none	1:100	1:10
Treatment		MP, PLEX, Rit.	MP, Rit.	MP, PLEX, Rit.	MP
Epitopes		KKPAK, HTFR, CEGPP,	GTSVVC, AQPFTGKCIF, VEKTFRNYDN	NDEYV, VEYL	ISEGSF, ISDO, CIITE, LND, PIVIET, VEY
HLA Type		DRB1*04:02[DR4]; DQA1*03:01;DQB1*03:02[DQ8(3);DPA1*01:03;DPA1*02:01,DPB1*01:01[DPw1],DPB1*04:01[DPw4]	DRB1*07:01[DR7],DQA1*02:01,DQB1*02:02[DQ2],DRB1*08:01[DR8], DQA1*04:01, DQB1*04:02[DQ4	DRB1*04[DR4],DQA1*03,DRB1*07[DR7],DQB1*02:02[DQ2],DQB1*03:02[DQ8(3),DPA1*01:03,DPA1*02:01,DPB1*02:01[DPw2],DPB1*04:01[DPw4]	DRB1*04:07[DR4],DQA1*02:01; DQB1*02:02[DQ2],DQB1*03:01[DQ7(3)],DRB1*01[DR7
Electrophysiology	LTP mean of averaged potentiation values (%)	Patient 1: 134.32% ± 3.92; Control: 128.06% ± 3.31	Patient 2: 118.88% ± 3.35; Control: 120.81% ± 3.63	Patient 3: 123.36% ± 3.37; Control: 124.80% ± 4.80	Patient 4: 132.9% ± 3.6; Control: 134.4% ± 8.4
	1 h incubation	*N* = 13, *n* = 27	*N* = 6, *n* = 12	*N* = 11, *n* = 19	*N* = 10, *n* = 18
		*p* = 0.419	*p* = 0.876	*p* = 0.706	*p* = 0.810
	6 h incubation	Patient 1: 111.67% ± 4.07; Control: 126.29% ± 3.94	Patient 2: 118.30% ± 3.92; Control: 115.34% ± 2.75	Patient 3: 115.38% ± 2.33; Control: 115.4% 8 ± 2.45	Patient 4: 120.3% ± 7.2; Control: 125.6% ± 6.9
		*N* = 13, *n* = 31	*N* = 6, *n* = 10	*N* = 10, *n* = 23	*N* = 8, *n* = 21
		* p * = 0.015	*p* = 0.601	*p* = 0.451	*p* = 0.978
Spine density	spine density	Patient 1: 1.91 ± 0.05; Control: 1.95 ± 0.05 spines/µm dendrite	Patient 2: 2.04 ± 0.06; Control: 1.96 ± 0.06 spines/µm dendrite	Patient 3: 1.86 ± 0.04; Control: 1.87 ± 0.03 spines/µm dendrite	Patient 4: 2.19 ± 0.04; Control: 2.18 ± 0.04 spines/µm dendrite
CA1 apical	1 h incubation	*N* = 3, *n* = 15	*N* = 3, *n* = 15	*N* = 3, *n* = 15	*N* = 3, *n* = 15
		3086 spines	5365 spines	3258 spines	2738 spines
		*p* = 0.548	*p* = 0.9013	*p* = 0.393	*p* = 0.202
	6 h incubation	Patient 1: 1.84 ± 0.04; Control: 1.78 ± 0.04 spines/µm dendrite	Patient 2: 2.00 ± 0.03; Control: 1.96 ± 0.04 spines/µm dendrite	Patient 3: 1.80 ± 0.04; Control: 1.87 ± 0.03 spines/µm dendrite	Patient 4: 2.07 ± 0.03; Control: 2.19 ± 0.03 spines/µm dendrite
		*N* = 3, *n* = 15	*N* = 3, *n* = 15	*N* = 3, *n* = 15	*N* = 3, *n* = 15
		3128 spines	4705 spines	3152 spines	3067 spines
		*p* = 0.345	* p * = 0.0152	*p* = 0.414	*p* = 0.147
Dentate Gyrus	6 h incubation	Patient 1: 1.73 ± 0.05; Control: 1.89 ± 0.03 spines/µm dendrite	Patient 2: 2.23 ± 0.07; Control: 2.19 ± 0.07 spines/µm dendrite	Patient 3: 2.00 ± 0.05; Control: 1.96 ± 0.06 spines/µm dendrite	Patient 4: 2.08 ± 0.05; Control: 1.25 ± 0.03 spines/µm dendrite
		*N* = 3, *n* = 15	*N* = 3, *n* = 15	*N* = 3, *n* = 15	*N*= 3, *n* = 15
		3137 spines	2819	3489	3310
		* p * = 0.014	* p * < 0.0001	*p* = 0.630	*p* = 0.705

## Data Availability

Data can be requested via email from the corresponding author: peter.koertvelyessy@charite.de.

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
