# Peer review of "Antibody Properties Associate with Clinical Phenotype in LGI1 Encephalitis"

_cells, 2023, doi:10.3390/cells12020282_

Round 1

Reviewer 1 Report

This is a study aiming to corelate between LGI1 phenotype and Ab target.

this is a very interesting question and may be valuable to the understanding of the pathophysiology of this disease.

The methods part is very detailed and well written and is a very good and solid base for a study like this. the problem is that the number of patients is very small to conclude from and the conclusions are not supported enough by the results or at least not in the way these are written now.

comments:

1. page2 line 47 - eventually some patients develop...

most patients if not treated develop encephalitis - cognitive impairment

2. Table 1:

There is no IgG subclass for LE#2

it says that FBDS#1 had complex partial seizures which is usually apart of encephalitis so I am not sure I would call this patient isolated FBDS patient. in the result section the patient is described as having piloerection and FBDS (piloerection is usually not complex partial unless having consciousness change..)

3. Table 1 - HLA - LE#1 and FBDS #1 - DP is presented and in the other 2 patients is not. why ?

4. table 1 - LTP mean - looks like there is no connection between the header and the data - mv/ms = ? n=? N=?

5. table 1 - spine density n=? N=? spine number compared to what ? p value between what ?

6. page 6 line 217 - says that LE patients had IgG3, table show LE1 and FBDS1...

7. table 1 - patient 2 doesn't have any IgG...

8. Figure1 presents LE#1, FBDS#2, FBDS#3 ???

9. In figure 1 FBDS patient has IgG3 in contrary to what is written...

10. in figure 1 legend it says - clinical data on four... - there is no clinical data

11. figure 2 and 3 are in the wrong place and hard to understand which text relate to which figure. the authors should mention above each figure the figure number.

12. page 7 line 238 - there is no #3

13. page 8 line 253 - what do the authors mean by wash-in of individual fractions ?

14. page 8 line 254 - says that there was no change after 1 or 6 hours. c' and b''' seem reduced compared to controls... significant ?

15. page 10 line 267 - what is AAK ?

16. page 8:

While patient LE#1 IgG altered long-term potentiation after 6 h, IgGs of patient LE#2 impaired short-term plasticity after 6 h (Figure 4G’’’)

in page 10 line 268 - it says - no difference after 6 hours despite lower PPF curve - the authors refers to this as a change in short term plasticity... is the curve change enough for that statement or is the final value...

17. page 10 line 276 - C shows small reduction, C''' show larger reduction but there is 1 p value...

18. line 276-281 the sentence is not clear.

19. figure 3 legends - not clear, If I understand correctly, spine density was reduced in CA1 after 6 hours in LE#2, and in LE#1 and 2 after 6 hours in dentate neurons

20. why did the authors test only LE 1 and 2 in figure 5and not FBDS patients ? it would be interesting to see staining differences (probably would not be..) and more interesting to see clustering of AMPA receptors and apoptosis

21. my major concern is the conclusions made in the discussion.

the first sentence is:

Our data show that epitopes and to some degree IgG subclass composition of LGI1-specific ab correlate to the symptoms of LGI1 AE. Long-term potentiation, short-term plasticity and hippocampal spine density were affected by LGI1 IgG from patients with LE but not from ab derived from the FBDS patients.

I dont think that the results in this study are significant and strong enough to say that.

first: this is a very small number of patients. second, even in this small group the patients antibodies did not have the same effect - 1 LE patient influenced LTP, 1 influenced spines in CA-1 and the other didn't, 1 patient with LE targeted only the LRR while the other also targeted EPTP.

the 2 FBDS should have been purely FBDS especially in such a small group. the fact that one had complex partial seizures (although cognitive assessment was normal) makes it possible that this is a patient with encephalitis...

a previous large study by Ramberger  et.al - "Distinctive binding properties of human monoclonal LGI1 autoantibodies determine pathogenic mechanisms" published in brain 2020 described 31 patients:

From 31 patients (28/31 with limbic encephalitis, including 12/31 with FBDS, 1/31 FBDS only, 1/31 Morvan’s syndrome, 1/31 neuromyotonia/pain). Across all individuals, serum and CSF showed similar levels of LRR and EPTP reactivities, with the exception of 2/11 CSFs that showed exclusive LRR or EPTP.

The mentioned study findings are in contrast to the findings in this study. the authors found that the 2 FBDS patients had antibodies targeting both LRR an EPTP domains and that LE patients targeted LRR (although not completely so...) and hypothesize that this is the reason, at least in part, to the different phenotype. the fact that a study of 31 patients of which 28 had limbic encephalitis found opposite results is not supporting this hypothesis.

I think the limitations should be addressed much more seriously as well as the differences I mentioned and the conclusions should be much more modest.        

Author Response

This is a study aiming to corelate between LGI1 phenotype and Ab target. this is a very interesting question and may be valuable to the understanding of the pathophysiology of this disease.

The methods part is very detailed and well written and is a very good and solid base for a study like this. the problem is that the number of patients is very small to conclude from and the conclusions are not supported enough by the results or at least not in the way these are written now.

We thank the reviewer for the highly appreciated comments. We will answer everything in a point by point reply.

We have highlighted every change made in the manuscript in the template version.

comments:

  1. page2 line 47 - eventually some patients develop...

most patients if not treated develop encephalitis - cognitive impairment

Done

  1. Table 1:

There is no IgG subclass for LE#2

We have changed the table1. One change is that we have deleted the IgG subclass analysis from table1 since figure 1 is showing all the subclasses.

  1. it says that FBDS#1 had complex partial seizures which is usually apart of encephalitis so I am not sure I would call this patient isolated FBDS patient. in the result section the patient is described as having piloerection and FBDS (piloerection is usually not complex partial unless having consciousness change..)

We changed to simple partial seizures after reviewing the complete record of the patient.

  1. Table 1 - HLA - LE#1 and FBDS #1 - DP is presented and in the other 2 patients is not. why ? We decided to share the entire HLA TypII sequencing results in Table 1 for better understanding.

  1. table 1 - LTP mean - looks like there is no connection between the header and the data - mv/ms = ? n=? N=?

changed

  1. table 1 - spine density n=? N=? spine number compared to what ? p value between what ?

Have defined the p value

  1. page 6 line 217 - says that LE patients had IgG3, table show LE1 and FBDS1...

We changed it accordingly, pleasew also pay attention to figure 1a

  1. table 1 - patient 2 doesn't have any IgG...

Patient 2 also known as FBDS#1 has IgG1-2 and predominantly IgG4. We changed the table1 and please pay attention to figure 1.

  1. Figure1 presents LE#1, FBDS#2, FBDS#3 ???

We corrected the Figure1 accordingly

  1. In figure 1 FBDS patient has IgG3 in contrary to what is written...

Only one FBDS patient has IgG3 it is the FBDS#2 patient.

  1. in figure 1 legend it says - clinical data on four... - there is no clinical data

We corrected the legend

  1. figure 2 and 3 are in the wrong place and hard to understand which text relate to which figure. the authors should mention above each figure the figure number.

We clarified everything in the text

  1. page 7 line 238 - there is no #3

Corrected

  1. page 8 line 253 - what do the authors mean by wash-in of individual fractions ?

At the beginning of each electrophysiological experiment there is a first phase of antibodies injction into the system called wash-in.

  1. page 8 line 254 - says that there was no change after 1 or 6 hours. c' and b''' seem reduced compared to controls... significant ?

Please have a look at the new table1 summarizing the significant effects.

  1. page 10 line 267 - what is AAK ?

We corrected the term

  1. page 8:

While patient LE#1 IgG altered long-term potentiation after 6 h, IgGs of patient LE#2 impaired short-term plasticity after 6 h (Figure 4G’’’)

in page 10 line 268 - it says - no difference after 6 hours despite lower PPF curve - the authors refers to this as a change in short term plasticity... is the curve change enough for that statement or is the final value...

Yes it is, because the effect appears at an interstimulus interval of 40ms and becomes significant at 20ms (p=0.049) and 10ms (p=0.049).

  1. page 10 line 276 - C shows small reduction, C''' show larger reduction but there is 1 p value...

We also wondered about that but the statistics is pretty clear.

  1. line 276-281 the sentence is not clear.

We corrected the sentence

  1. figure 3 legends - not clear, If I understand correctly, spine density was reduced in CA1 after 6 hours in LE#2, and in LE#1 and 2 after 6 hours in dentate neurons

You understood it right

  1. why did the authors test only LE 1 and 2 in figure 5and not FBDS patients ? it would be interesting to see staining differences (probably would not be..) and more interesting to see clustering of AMPA receptors and apoptosis

We did not do the laborious and elaborate staining of the hippocampus because we did not see any electrophysiological hint for a change in AMPA receptors (basal synaptic transmission) nor in NMDA receptors (long-term-potentiation). Staining hippocampus slides after one year in the freezers did not work out so far due to tissue changes. We learned this from another experiments. Actually, these experiments were performed 2.5 years ago

  1. my major concern is the conclusions made in the discussion.

the first sentence is:

Our data show that epitopes and to some degree IgG subclass composition of LGI1-specific ab correlate to the symptoms of LGI1 AE. Long-term potentiation, short-term plasticity and hippocampal spine density were affected by LGI1 IgG from patients with LE but not from ab derived from the FBDS patients.

I dont think that the results in this study are significant and strong enough to say that.

first: this is a very small number of patients. second, even in this small group the patients antibodies did not have the same effect - 1 LE patient influenced LTP, 1 influenced spines in CA-1 and the other didn't, 1 patient with LE targeted only the LRR while the other also targeted EPTP.

The 2 FBDS should have been purely FBDS especially in such a small group. the fact that one had complex partial seizures (although cognitive assessment was normal) makes it possible that this is a patient with encephalitis...

Dear reviewer, we changed almost the entire section on discussion and conclusion being more modest in our discussion. We have highlighted the entire section to emphasize the numerous changes made. We hope to dispel your concerns.

A previous large study by Ramberger  et.al - "Distinctive binding properties of human monoclonal LGI1 autoantibodies determine pathogenic mechanisms" published in brain 2020 described 31 patients:

From 31 patients (28/31 with limbic encephalitis, including 12/31 with FBDS, 1/31 FBDS only, 1/31 Morvan’s syndrome, 1/31 neuromyotonia/pain). Across all individuals, serum and CSF showed similar levels of LRR and EPTP reactivities, with the exception of 2/11 CSFs that showed exclusive LRR or EPTP.

The mentioned study findings are in contrast to the findings in this study. the authors found that the 2 FBDS patients had antibodies targeting both LRR an EPTP domains and that LE patients targeted LRR (although not completely so...) and hypothesize that this is the reason, at least in part, to the different phenotype. the fact that a study of 31 patients of which 28 had limbic encephalitis found opposite results is not supporting this hypothesis.

We added a more detailed discussion of the results by Ramberger et al. in the discussion session.

Ramberger et al abstract: “Taken together, two largely dichotomous populations of LGI1 mAbs with distinct domain binding characteristics exist in the affinity matured peripheral autoantigen-specific memory pools of individuals, both of which have pathogenic potential. In human autoantibody-mediated diseases, the detailed characterization of patient mAbs provides a valuable method to dissect the molecular mechanisms within polyclonal populations.“

We cannot see a big difference to the Ramberger et al. paper as our reviewer. I just copied the last sentences of the abstract of this sentinel paper supporting our results.

Ramberger et al. only did polyclonal IgG purification from 4 patients as we did (page 1733 bottom left) and sequenced monoclonal antibodies from only two patients. They performed electrophysiological measurements with these two monoclonal antibodies and we did electrophysiologcal studies with 4 patients antibodies. We do added some data to this

The main difference between our studies, is a highly specific epitope mapping in our study compared to the Ramberger et al, study, allowing us to precisely localize the epitopes of each antibody on the LGI1 Protein taking into account the tertiar structure. Ramberger et al. used whole length EPTP and LRR and whole-length LGI1 protein. Our localization potential gives us more insight to where the antibodies bind on LGI1 showing us no overlapping epitopes at all (Figure1 B). We believe that the precise localization for example inside or outside the beta sheet make a difference on the effect oft the antibody.

They focused more on other aspects in their excellent paper than combining individual symptoms with extensive electrophysiology and precise epitope mapping. So, we think that we are not in total contrast to them but support and replenish this work.

This huge difference is also mirrored by the electrophysiological results as we pointed out. We do see the problem that the more precise you get the bigger your cohort must be, what we unfortunately cannot offer.

In sum, there is a difference between the patients epitope and electrophysiological which noone else could detect before due to there limited knowledge about the epitopes.

We reworte the discussion and conclusion parts and made it more modest and discussed the Ramberger et al. Paper in more detail.

I think the limitations should be addressed much more seriously as well as the differences I mentioned and the conclusions should be much more modest.       

Reviewer 2 Report

Ludewig et al performed an autoantibody characterization in LGI1 encephalitis.

Although it is an interesting approach, I find that the article's conclusions are presented with a high degree of certitude not justified by the evidence (only 4 patients, 2 different phenotypes, incomplete data)

It would be interesting to see for the patients where the antibodies persisted after the treatment if the proportion between the antibody isotype is maintained.

Author Response

Ludewig et al performed an autoantibody characterization in LGI1 encephalitis.

Although it is an interesting approach, I find that the article's conclusions are presented with a high degree of certitude not justified by the evidence (only 4 patients, 2 different phenotypes, incomplete data)

We acknowledge this major point and rephrased almost the entire discussion and conclusion section

It would be interesting to see for the patients where the antibodies persisted after the treatment if the proportion between the antibody isotype is maintained.

This is indeed a very interesting question leading us to a next experiment. We added it to our last sentence of the manuscript: “In further studies a higher number of patients as well as other experimental setups are needed to evaluate the anti-LGI1 IgG subclasses contribution and epitope spreading to anti-LGI1 ab-mediated AE over the course of the disease.”

Sadly, the answer to this interesting is beyond the scope of the experiments performed here, so we provided an extensive method section in case anyone else wants to go further on in this direction. We do have serial serum samples and should go for more funding.

Round 2

Reviewer 1 Report

After the corrections made I still think the number of patients is too small for the conclusions and the study itself.

On the other hand, taking all into consideration, I rather accept than not, since we can still learn from the study.